# Basil (*Ocimum basilicum* L.) Leaves as a Source of Bioactive Compounds

**DOI:** 10.3390/foods11203212

**Published:** 2022-10-14

**Authors:** Raffaele Romano, Lucia De Luca, Alessandra Aiello, Raffaele Pagano, Prospero Di Pierro, Fabiana Pizzolongo, Paolo Masi

**Affiliations:** 1Department of Agricultural Sciences, University of Naples Federico II, Via Università, 100, 80055 Portici, NA, Italy; 2CAISIAL, University of Naples Federico II, Via Università, 133, 80055 Portici, NA, Italy

**Keywords:** supercritical carbon dioxide, polyphenols, linalool, caffeic acid, antioxidant activity, volatile organic compounds, green technology

## Abstract

Basil (*Ocimum basilicum* L.) is an annual spicy plant generally utilized as a flavouring agent for food. Basil leaves also have pharmaceutical properties due to the presence of polyphenols, phenolic acids, and flavonoids. In this work, carbon dioxide was employed to extract bioactive compounds from basil leaves. Extraction with supercritical CO_2_ (*p* = 30 MPa; T = 50 °C) for 2 h using 10% ethanol as a cosolvent was the most efficient method, with a yield similar to that of the control (100% ethanol) and was applied to two basil cultivars: “Italiano Classico” and “Genovese”. Antioxidant activity, phenolic acid content, and volatile organic compounds were determined in the extracts obtained by this method. In both cultivars, the supercritical CO_2_ extracts showed antiradical activity (ABTS^●+^ assay), caffeic acid (1.69–1.92 mg/g), linalool (35–27%), and bergamotene (11–14%) contents significantly higher than those of the control. The polyphenol content and antiradical activity measured by the three assays were higher in the “Genovese” cultivar than in the “Italiano Classico” cultivar, while the linalool content was higher (35.08%) in the “Italiano Classico” cultivar. Supercritical CO_2_ not only allowed us to obtain extracts rich in bioactive compounds in an environmentally friendly way but also reduced ethanol consumption.

## 1. Introduction

Basil (*Ocimum basilicum *L.), belonging to the *Lamiaceae* family, is an annual spicy plant generally utilized as a flavouring agent for food. Although basil is indigenous to India, it is cultivated worldwide and is particularly appreciated in cuisine for its distinctive aroma (given by essential oils). Basil is mainly used as a spice but is also considered a main ingredient in certain preparations, such as “Pesto Genovese” [1]. In 2017, in Italy, according to statistical data [2], 27,132 hectares of protected cultivation were cultivated for a production of 8908.9 metric tons of basil. Liguria represents the first Italian region in basil production and is known for “Genovese” basil, a protected designation of origin (PDO) product essential in the success of “Pesto Genovese”. The world market for basil is growing, and in recent years, exports to the USA, Great Britain, France, and Germany have grown. In addition to its use as a spice and as an ingredient in food preparations, basil is also valuable for its cosmetic and pharmaceutical properties due to its large amounts of essential oils and phenolic compounds, particularly rosmarinic acid [3].

Basil is composed, on average, of approximately 92.1% water, 3.15% proteins, and 2.65% carbohydrates, of which 0.3% are soluble sugars, 1.6% are fibre, 0.75% are other carbohydrates, 0.64% are total lipids, and 1.49% are ash [2]. The phenolic compounds in basil are primarily phenolic acids such as rosmarinic, chicoric, ferulic, and caffeic acid [4,5]. Several studies reported that due to the synergistic antioxidant effect of these compounds, basil extract is a good source of natural antioxidants [6] with health-benefit activities such as antimicrobial, anti-inflammatory, and antidiabetic properties [3,7].

Phenolic compounds reduce the risk of cardiovascular and degenerative diseases by preventing oxidative stress and the oxidation of biological macromolecules. In fact, they scavenge free radicals and possess metal-chelating properties. Moreover, they have anticancer activities [8,9,10]. Additionally, consumption of foods rich in phenolic compounds is associated with various physiological effects, such as preventing cancer and certain chronic diseases [11].

Rosmarinic acid is one of the most abundant polyphenols present in basil [4] and is known for its great antioxidant activity. This compound is used for the treatment of ulcers, arthritis, cataracts, cancer, and other illnesses because of its antimicrobial, anti-inflammatory, anti-mutagenic, anti-viral, and anticancer activities in addition to its antioxidant properties [12]. Furthermore, in the food industry, rosmarinic acid is utilized as a food preservative to extend the shelf life of fresh seafood [13].

The presence of caffeic, chicoric, and ferulic acid has also been reported in basil [4]. Caffeic acid is a constituent of hydroxycinnamic acid, which is a very common dietary ingredient for humans. This acid has different biological activities, such as antibacterial activity, antioxidant properties, antiatherosclerotic activity, immunostimulant activity, anti-inflammatory/analgesic activity, antiviral activity, and cardioprotective and antiproliferative activity [14].

Chicoric acid has main benefits, such as antiviral, anti-inflammatory, glucose and lipid homeostasis, neuroprotective, and antioxidative effects [15].

Ferulic acid has different functions, such as antioxidant, anti-inflammatory, antimicrobial, antiallergic, anticarcinogenic, and antiviral activities, the capacity to increase serum viability, vasodilatory actions, metal chelation, modulation of enzyme activities, activation of transcription factors, gene expression and signal transduction in biological systems, and the capacity to inhibit lipid peroxidation [16,17].

The beneficial health properties of basil are also attributed in large part to its essential oil. Essential oils derive from the secondary metabolism of plants and are composed of complex mixtures of low molecular weight compounds [18]. They could be obtained from different methods of extraction and applied to extend the shelf life of foods, owing to their antimicrobial and antioxidant properties, in addition to their use as flavouring agents [19]. Overall, the principal compounds of basil essential oil are terpenes and phenylpropanoids, followed by alcohols and aldehydes [20]. According to several studies [21], biological activities of the essential oil from *O. basilicum* is due mainly to linalool and phenolic compounds. These activities include antimicrobial, antifungal, insect repelling, antioxidant, anticancer, and anti-inflammatory activities.

The extraction of the various bioactive compounds with beneficial properties is often performed with conventional extraction techniques that are generally based on the use of various solvents, such as ethanol, methanol, hexane, etc., using heating or mixing. The conventional extraction techniques have several disadvantages including long extraction times, the need for expensive purity solvents, low selectivity, evaporation of significant amounts of solvents, and potential decomposition of thermolabile compounds [22,23]. These problems can be solved by using other extraction techniques, such as green techniques. In particular, the use of supercritical fluids in the production of volatile oils and extracts has increased since 2000 due to its advantages, such as simplicity, rapidity, and selectivity. These techniques have received increased attention for the production of plant extracts for the pharmaceutical, cosmetics, and food industries [24,25]. Carbon dioxide (CO_2_) is a nonpolar solvent that can serve as a supercritical fluid using low temperature and intermediate pressure values. It is common practice to improve the solvating power of supercritical CO_2_ by small additions of organic modifiers (cosolvents) of differing polarity to include more polar compounds [26]. This is an environmentally friendly extraction technique because it reduces the use of organic solvents and is in line with the principle of sustainability. In fact, CO_2_ is a nontoxic, nonflammable, recyclable, and inexpensive solvent that is easily removed from extracts. This technique has already been used to extract polyphenols from other vegetable matrices [26,27,28,29] with excellent results in terms of yield and quantity of extracted bioactive compounds. In a recent work [27], we have already used CO_2_ in liquid and supercritical conditions to extract oils from hemp seed with a large amount polyphenols and tocopherols. We also used liquid and supercritical CO_2_ to produce oleoresins with high quantities of polyphenols and with important antioxidant activity from tomato waste [29] as well as to recover bioactive compounds from walnut green husk and citrus peels [26,28].

In this work, we used the CO_2_ in supercritical and liquid phase to produce extract from basil leaves. The use of ethanol as a cosolvent was proposed. Parameters such as time and percentage of ethanol were varied to increase the yield. The conventional extraction method with pure ethanol was used as the control. Subsequently, two basil cultivars, “Italiano Classico” and “Genovese”, were submitted to the selected supercritical CO_2_ extraction, and the extracts were characterized in terms of phenolic compounds, volatile organic compounds, and antioxidant activities.

## 2. Materials and Methods

### 2.1. Materials

The plant material used in this study consisted of leaves collected from the plants of two cultivars (“Genovese” and “Italiano Classico”) of common basil (*Ocimum basilicum* L.) purchased from the Campania district area.

Basil leaves were frozen at −18 °C and lyophilized at −50 °C, <0.05 mbar for 48 h. The samples were stored in the dark at −18 °C until the extractions.

### 2.2. Chemicals

All solvents and reagents for the experiments were purchased from Merck, (Darmstadt, Germany). The carbon dioxide (CO_2_) (assay purity 99.9%) was provided by SOL Spa (Naples, Italy).

### 2.3. Moisture Content

The moisture content was gravimetrically determined. Approximately 3 g of basil leaves were weighed before and after lyophilization performed at −50 °C, <0.05 mbar for 48 h. The results were expressed as a weight/weight percentage of water (% *w*/*w*).

### 2.4. Organic Solvent Extraction (Control)

Lyophilized basil leaves were ground and sieved to obtain particles with ø ≤ 1 mm according to Romano et al. [28]. A quantity of 15 mL of ethanol was added to approximately 400 mg of dried and ground material and the extraction was performed according to Romano et al. [28] for 1.5 h. The extraction was repeated 4 times, consuming a total of 60 mL of ethanol for 400 mg of leaves. The extraction method used as the control was coded C (Table 1). The extraction yields were expressed as g extract/100 g dry matter (DM). The extracts were stored at −20 °C until analysis.

### 2.5. Supercritical and Liquid CO_2_ Extraction

The CO_2_ extraction was performed according to Romano et al. [28] with some modifications. Approximately 4 g of dried and ground material (ø ≤ 1 mm) was added to an SFC 4000 extractor (JASCO International Co., Ltd., Tokyo, Japan) equipped with a 50 mL volume extractor vessel.

Supercritical (SC) and liquid (L) CO_2_ extractions were performed at different flow rates, temperatures, pressures, times, and percentages of ethanol used as cosolvents. The extraction methods were coded as shown in Table 1. During each type of extraction, 30 min of the static phase was alternated with 30 min of the dynamic phase. The alternation of static and dynamic process improved the SC extraction recovery due to the equilibrating effect on the solute transfer from solid particles to fluid phase [30]. Furthermore, the static phase is necessary for stabilization of temperature in the extraction vessel as well as for obtaining a good penetration of the solvent comprising ethanol and CO_2_ into the matrix [31]. The extraction was complete after 5 h with all percentages of ethanol used, both in SC and L CO_2_ extractions (in SC1, SC2, SC3, L1, L2, an dL3). It was shortened to 2 h (in SC4, SC5, SC6, L4, L5, and L6) and to 1 h (in SC7, SC8, SC9, L7, L8, and L9) to verify whether similar yield values can be obtained by shortening the extraction.

The extraction yields were expressed as g extract/100 g DM. The extracts were stored at −20 °C until analysis.

### 2.6. Total Polyphenol Content

The Folin–Ciocalteu method reported by Ahmed et al. [32] with modifications was used to determine the total polyphenol content (TPC) of extracts. Briefly, 10 mg of extract was added to 10 mL of methanol and shaken for 2 min. To 0.2 mL of this solution, 2.5 mL of 10% Folin–Ciocalteu reagent diluted in water was added. After incubation for 2 min at ambient temperature in the dark, 2 mL of a 7.5% sodium carbonate solution was added. After another 120 min of incubation at ambient temperature in the dark, the absorbance was measured at 765 nm using a UV-1601PC UV—Visible spectrophotometer (Shimadzu, Milan, Italy). A calibration curve (R^2^ = 0.99) was constructed with gallic acid at different concentrations (25, 50, 100, 150, and 200 mg/L). The results were expressed as mg of gallic acid equivalent/g of extract (mg GAE/g).

### 2.7. Individual Polyphenols by High-Performance Liquid Chromatography (HPLC) Analysis

The individual polyphenols were determined by high performance liquid chromatography (HPLC) analysis according to Ciriello et al. [33]. A quantity of 100 mg of extract was dissolved with 2 mL of 70% methanol–water. The mixture was filtered with a 0.45 µm polyethersulfone (PES) filter (Phenomenex, Torrance, CA, USA) and 20 μL was injected into the HPLC system (Agilent 1100 Series, Santa Clara, CA, USA). A reversed-phase C18 column (150 × 4.6 mm i.d.; particle size 5 μm; Kinetex^®^ 100 Å; Phenomenex, Torrance, CA, USA) was employed. The mobile phases were 0.1% trichloroacetic acid in water (phase A) and acetonitrile (B), and the elution gradient was 0–50 min and 50% B. The flow was set to 1.0 mL/min. Detection was performed at 280 and 303.6 nm. To quantify the concentrations of compounds, calibration curves of standards (caffeic acid, rosmarinic acid, ferulic acid, and chicoric acid) were constructed. The range of linearity was 10–100 ppm for all standards, and the square of the correlation coefficient (*R*^2^) was ≥0.99. The limit of detection (LOD) and limit of quantification (LOQ) were 2.5 and 5 ppm, respectively, for all compounds. The results were expressed as mg of the phenolic compound/g of extract.

### 2.8. Antioxidant Activity Assays

The antioxidant activity of the extracts was determined by DPPH^●+^, ABTS^●+^, and FRAP assays.

The DPPH^●+^ and ABTS^●+^ assays are based on the scavenging activity of free radical.

The 2,2-diphenyl-1-picrylhydrazyl (DPPH^●+^) was performed as described by Coelho et al. [32] with certain modifications. Briefly, the extracts were dissolved in methanol at known concentrations between 0.2 and 3.2 mg/mL. A quantity of 30 μL of diluted extract was added to 270 μL of 100 μM DPPH^●+^ in methanol. After incubation in the dark for 60 min at room temperature, the absorbance was measured at 517 nm using a UV-1601PC UV—Visible spectrophotometer (Shimadzu, Milan, Italy). The DPPH^●+^ scavenging effect was expressed as a percentage of discolouration of DPPH^●+^ compared with a control solution:% scavenging effect = [(Ac − As)/(Ac)] × 100 (1)
where Ac is the control absorbance and As is the sample absorbance.

The EC_50_ values (mg/mL of extract), which is the concentration that allows 50% reduction of a solution with a known titre of DPPH^●+^, were then determined.

The 2,2′-azinobis-3-ethyl-benzothiazoline-6-sulfonic acid (ABTS^●+^) assay was performed as described by Coelho et al. [34] with certain modifications. The extracts were dissolved in methanol at known concentrations between 0.2 and 3.2 mg/mL. A quantity of 20 μL of diluted extract was added to 280 μL of ABTS^●+^ working solution. The solution was kept in the dark for 5 min, and the absorbance was measured at 734 nm using a spectrophotometer. The percentage of discolouration of ABTS^●+^ was calculated as performed in the DPPH^●+^ assay, and the antioxidant activity was expressed as EC_50_ values (mg/mL of extract).

The ferric reducing/antioxidant power (FRAP) procedure was conducted according to the modified method of Kwee and Niemeyer [35]. An aliquot of 70 μL of diluted extract was added to 279 μL of FRAP reagent and 651 μL of acetate buffer. The Trolox calibration curve was prepared in the range of 25–400 µM. After 4 min of incubation at ambient temperature in the dark, the absorbance was recorded at 593 nm. The results were expressed in mmol Trolox equivalents (TE)/g of extract.

### 2.9. Volatile Organic Compounds

The analysis of volatile organic compounds (VOCs) in the extracts was performed using the solid phase microextraction technique (SPME) coupled with gas chromatography according to Ciriello et al. [33] with certain modifications. In particular, 50 mg of extract was weighed in a 10 mL vial for headspace analysis. Other details on methodologies, instrumentations, and peak identification were reported in Romano et al. [28]. The relative content of VOCs was calculated from peak area ratios and expressed in terms of percentage (%).

### 2.10. Statistical Analysis

All experiments were performed in triplicate, and the results are expressed as the mean values (±standard deviations) of the three replicates. The data were submitted to one-way analysis of variance (ANOVA) and Tukey’s multiple-range test (*p* ≤ 0.05) using XLSTAT software (Addinsoft, New York, NY, USA).

## 3. Results and Discussion

### 3.1. Choice of the Extraction Method

In Table 1, the extraction yields for the different methods used are reported. The control method with ethanol showed the highest yield of 7.98 g/100 g DM. The SC3 and SC6 methods showed yield values near those of the control (6.54 and 6.26 g/100 g DM, respectively). Despite the yield difference of plus 1.4, the SC3 and SC6 methods were more convenient than C because they allow for a reduction in the extraction time, the quantity of ethanol, and the costs. In fact, ethanol, especially food grade, is more expensive than CO_2_. Moreover, it is preferable to use low percentage of ethanol to obtain low/free solvent products and to reduce the use of organic solvents. The application of organic solvents and their negative influence on the environment is a problem for health, and their use has been recognized to be a matter of important environmental concern [36].

Between the SC3 and SC6 methods, the SC6 was chosen to produce the extracts from the two cultivars (“Genovese” and “Italiano Classico”) of basil for shortness (2 h versus 5 h) and because the yield was not significantly different (*p* ≤ 0.05) from SC3. The SC6 method reduces energy and solvent consumption. Additional extractions were also carried out by using the same parameters of SC3 and SC6 methods with 20% of ethanol as cosolvent in order to verify whether the yield could be further increased. However, the results (6.60 and 6.31 g/100 g DM, respectively) were not significantly different from the yields obtained with SC6 method.

All SC methods showed higher yields compared with L methods under the same conditions of time, flow, and percentage of cosolvent. This finding is attributed to the lower ability of liquid CO_2_ to penetrate the matrix compared with CO_2_ in the supercritical state [27]. Furthermore, the decrease in extraction pressure in the L methods determines a decrease in CO_2_ density and a yield that is lower than that of the SC methods [37]. Additionally, according to Elgndi et al. [38], who carried out CO_2_ supercritical extractions on basil at 100 and 300 bars at a constant temperature, the increase in pressure is directly related to the increase in the dissolving capacity of CO_2_. For both the SC and L methods, the use of 0% ethanol produced yields significantly (*p* ≤ 0.05) lower than the equivalent with 5 and 10% ethanol. SC4, SC7, L4, and L7 showed yields < 0.10%. The use of 10% ethanol caused an increase in the extraction yield, as a polar solvent such as ethanol increases the polarity during extraction [29]. Notably, a common practice, especially in the extraction of supercritical fluids, is to change the polarity of the fluid and to increase the solvent power towards the analyte of interest using polar modifiers (cosolvents) [29]. Modifiers can also reduce analyte–matrix interactions, improving their quantitative extraction [39].

### 3.2. Polyphenol Content

The total polyphenol content (TPC) in SC6 extracts from the two cultivars compared with the control is shown in Table 2. The SC6 extracts showed 72.15 and 79.62 mg GAE/g in “Italiano Classico” and “Genovese”, respectively, while the control method obtained values of 91.66 and 98.99 mg GAE/g, respectively. The same trend was also observed by Coelho et al. [34] who reported higher values in ethanolic extract (70–85 mg GAE/g) compared with those in samples obtained by supercritical CO_2_ (35 mg GAE/g). This finding can be explained by the polarity of ethanol that allows the extraction of polar compounds such as polyphenols. The higher values obtained in our extracts are explained by the use of 10% ethanol as a cosolvent, unlike in the analysis of Coelho et al. [34]. In both types of extraction (SC6 and C), “Genovese” showed more TPC than the “Italiano Classico” cultivar.

The main (>0.1%) individual polyphenols detected in basil were rosmarinic, caffeic, chicoric, and ferulic acids (Table 2). The sum of individual polyphenols ranged from 8.20 to 10.78 mg/g extract and showed the same trend of TPC. It was higher in “Genovese” than in the “Italiano Classico” cultivar in both types of extraction (SC6 and C), and it was higher in C (100% ethanol) than in the SC6 extraction method in both cultivars.

Rosmarinic acid was the most represented phenolic acid in all types of extracts, with values in the range of 5.79–7.48 mg/g extract in SC6 samples, which were similar to those obtained in C samples (8.08–8.76 mg/g extract). These findings showed agreement with the literature [3,40,41,42], where rosmarinic acid is reported to be the most represented phenolic acid in basil. “Genovese” showed a value (7.48 mg/g extract) significantly higher than that of the “Italiano classico” cultivar. Numerous biological properties of rosmarinic acid have been described in particular antimicrobial, antidepressant, cytoprotective, antiviral, antiallergic, antiangiogenic, and antitumor activities [43,44,45,46,47].

The SC6 extracts showed values of chicoric and ferulic acid in the range of 0.49–0.54 mg/g extract and of 0.13–0.15 mg/g extract, respectively. These values were very similar to those of the ethanolic extracts, albeit slightly lower. Due to the similar polarity of these compounds and ethanol, the use of 10% ethanol as a cosolvent in the SC method allowed the recovery of a quantity of chicoric and ferulic acid very similar to that of the control. No significant (*p* ≤ 0.05) differences between the two cultivars were detected. Additionally, Lee and Scagel [4] detected chicoric acid in basil leaves with the highest level of 88.5 mg/100 g fresh weight. Both chicoric acid levels and ferulic acid levels reported by Bajomo et al. [48] ranged from 0.17 to 1.42 dry matter and from 0.08 to 0.58 mg/100 g dry matter, respectively, which were lower than our results. Discrepancies among basil phenolic acid concentrations reported in the literature may occur because of different postharvest and agronomic conditions and different cultivars.

The opposite trend was observed for the caffeic acid content, which was significantly (*p* < 0.05) higher in the SC6 extracts (1.69–1.92 mg/g extract) than in the control (0.97–1.24 mg/g extract). No significant (*p* ≤ 0.05) differences between the two cultivars were observed. Bitencourt et al. [49] discovered that by using 10.2% ethanol in SC CO_2_, caffeic acid solubility increased approximately 30,000-fold at 313 K and 20 MPa with respect to its solubility in pure SC CO_2_. In addition to its antioxidant activity, caffeic acid also exhibits anti-ischaemic, anti-thrombotic, anti-hypertensive, anti-fibrosis, anti-virus, and antitumour properties [49].

The production of natural extracts rich In phenolic compounds is interesting for the food, chemical, and pharmaceutical industries.

### 3.3. Antioxidant Activity

The antioxidant capacities of extracts need to be measured by more than one type of assay to take into account the various modes of action of antioxidants [26]; therefore, DPPH^●+^, ABTS^●+^, and FRAP assays were performed (Table 2). The lowest EC50 values (in the DPPH and ABTS assays) represent the highest antioxidant capacity, and vice versa. In both cultivars, the SC CO_2_ extract showed slightly lower antioxidant activity in the DPPH^●+^ test and FRAP test (1.86–1.52 mg/mL and 0.75–0.80 mmolTE/g, respectively) than the ethanolic extract (1.32–1.14 mg/mL and 0.88–0.92 mmolTE/g, respectively). The same trend, with comparable results, was also observed by Coelho et al. [34], who discovered in the ethanolic extract, a higher antioxidant capacity for the DPPH test results (IC_50_ = 3.99 mg/mL) and the reducing power test assay (285.1 TE/g extract) compared with supercritical fluid extracts (5.63 mg/mL and 111.7 TE/g, respectively). In our case, the differences between the two types of extraction in both assays were minimal, probably due to the use of ethanol as a cosolvent in the SC extraction, which allowed a greater recovery of the phenolic compounds, while Coelho et al. [34] performed SC CO_2_ extraction without cosolvent. It is well known that phenolic compounds have antioxidant capacity; notably, the correlation calculated between TPC and DPPH^●+^ and between FRAP and ABTS^●+^ was 0.977.

In all samples, the antioxidant activity measured by ABTS^●+^ was higher than that measured by the DPPH^●+^ assay (Table 2). Floegel et al. [50] verified that on a wide range of food matrices, the values of the ABTS^●+^ assay are often higher than those of the DPPH^●+^ assay, demonstrating that the ABTS^●+^ assay reacts with more components and in different ways from the DPPH^●+^ assay. For the ABTS^●+^ assay, the trend was opposite to the previous assays, showing agreement with Coelho et al. [34], and the antioxidant activity in SC CO_2_ extracts (0.842–0.733 mg/mL) was significantly higher (*p* < 0.05) than that in the ethanolic extract (1.294–1.115 mg/mL). This finding can be explained by the notion that the compounds responsible for the antioxidant activity are not necessarily the same for each extraction method [32]. For this test, the contribution of nonpolar compounds is probably important; notably, Coelho et al. [34] discovered that the antioxidant activity was approximately three times greater when extracted by SC CO_2_ without cosolvent compared with the ethanolic extract.

The antiradical activity measured by the three assays was higher in the “Genovese” cultivar extracts than in the “Italiano Classico” cultivar extracts.

### 3.4. Volatile Organic Compounds (VOCs)

The percentage composition of the volatile compounds of the extracts is shown in Table 3. The most characteristic compounds of basil leaves belong to the class of terpenoids, which are divided into monoterpenoids and sesquiterpenoids. The compounds detected in greater quantities were, in decreasing order, linalool, eugenol, trans-α-bergamotene, and eucalyptol for both extracts.

The linalool concentration ranged from 23.29 to 35.08%. Similar values were obtained by Telci et al. [51] in essential oil from different Turkish basil cultivars (from 4.3 to 48.3%), by Arranz et al. [52] in essential oil obtained by supercritical extraction (27.8%), and by Beatović et al. [53] in essential oil from different basil cultivars (from 11.5 to 58.6%). Lower values, instead, were reported by Rana and Blazquez [54], who discovered a maximum concentration of 16.7% in essential oil obtained with hydrodistillation, and by Coelho et al. [34], who obtained a concentration of 12.6% in essential oil extracted by supercritical extraction. Moreover, the linalool content was lower than that of Milenković et al. [20], who showed a concentration ranging from 42.3 to 47.2% in basil essential oils obtained by hydrodistillation.

Eugenol was detected in the range of 25.42–38.92%, which is higher than that reported by Milenković et al. [20], who obtained a maximum concentration of 20.9% depending on the growing conditions of basil used to prepare essential oils. Maggio et al. [55] reported eugenol concentrations ranging from 25 to 76% in basil essential oil depending on the *cultivar* selected for preparation, while Telci et al. [51] reported eugenol concentrations ranging from 3.1 to 21.1% depending on the basil chemotype.

Eugenol and linalool have an opposite trend; notably, when the eugenol concentration increased, the linalool concentration decreased, and vice versa, as reported by Said-Al Ahl et al. [56].

Eucalyptol was detected in the range of 1.64–5.12%, which is lower than that reported by Złotek et al. [57], who obtained a concentration of 16.68% in basil essential oils obtained by the hydrodistillation method, while the values were in the range reported by Marotti et al. [58], who discovered a concentration of 0.94–12.91% in essential oil obtained by hydrodistillation as a function of the analysed cultivar. The eucalyptol content was similar to the findings of Coelho et al. [34], who reported a concentration of 5.8% in basil essential oil extracted by supercritical extraction, and of Arranz et al. [52], who reported a concentration of 5.75% in basil essential oil obtained by supercritical extraction. Wei and Shibamoto [59] obtained a eucalyptol concentration of 2.2% in commercial essential oils, while Al-Abbasy et al. [60] discovered a eucalyptol concentration of 5% in essential oil extracted by hydrodistillation.

In all extracts, methyl-isoeugenol was also detected at low concentrations (1.02–2.56%). Krüger et al. [61] reported the presence of this compound in basil essential oil at a maximum concentration of 36%.

Notably, the composition of these compounds is also influenced by the type and season of cultivation, as well as by the variety of basil. The compositional difference is also attributed to the part of the plant used in oil extraction, to the country/region of collection, and to the selected extraction method [19]. Moreover, different physico-chemical changes in aromatic volatiles may occur during the drying process, influencing aroma intensity and the quality of the dried product [62].

Comparing the SC CO_2_ extraction with the control, some differences were observed in both cultivars (Table 3).

The linalool content was greater in the SC CO_2_ extracts (27.17–35.08%) than in the control (23.29–31.51%). Additionally, Occhipinti et al. [63] reported that extraction by CO_2_ increases the percentage of linalool compared with extraction by hydrodistillation. Linalool is the main component of basil essential oil [19]. Several studies have reported the numerous bioactive properties of linalool: inducing cycle arrest of human prostate cancer cells [64], inducing apoptosis in the myeloid leukaemia cell line [65], antimicrobial effects through the rupture of cell membranes, and protective effects on the liver, kidneys, and lungs due to its anti-inflammatory activity [66]. Due to its protective effects and low user toxicity, linalool can be employed as an adjuvant to anticancer drugs or antibiotics; therefore, it has great potential for use as a natural and safe therapeutic alternative [66].

Additionally, trans-α-bergamotene was detected in greater quantities in the SC CO_2_ extracts (11.08–14.40%) than in the control, as reported by Coelho et al. [34], who compared essential oils obtained by supercritical extraction and hydrodistillation. No significant differences were shown for eugenol, which also has numerous beneficial properties. The wide range of eugenol activities includes antimicrobials, anti-inflammatories, analgesics, and antioxidants [67,68].

Eucalyptol was detected in greater quantity in the SC CO_2_ extract than in the control only in the “*Genovese*” cultivar. Coelho et al. [34] also showed a higher concentration of this compound in extracts obtained by supercritical extraction. Due to its pleasant aroma and spicy taste, eucalyptol is utilized as a flavouring agent, fragrance, and in cosmetics. Furthermore, this compound has antinociceptive properties, anti-inflammatory, gastroprotective, and hepatoprotective effects, and antimycotic and antibacterial activity [69].

Differences between the two *cultivars* were identified: “Italiano Classico” showed higher linalool content (35.08%) and “Genovese” showed higher eugenol, trans-α-bergamotene, and eucalyptol contents (35.78, 14.40, and 5.12%, respectively).

## 4. Conclusions

The extraction by supercritical CO_2_ with 10% ethanol as cosolvent for 2 h from basil (*Ocimum basilicum* L.) leaves allowed us to obtain yields of approximately 6.26 g/100 g, which is very close to the yield of the control method carried out with 100% ethanol. Increasing the quantity of cosolvent to more than 10% did not produce significantly different yield. With the use of ethanol as cosolvent, the polarity of the extracting fluid was changed, expanding the amount of recoverable compounds. Nevertheless, it is preferable not to use a high percentage of ethanol to reduce the use of organic solvents and the costs.

The two basil cultivars, “Genovese” and “Italiano Classico”, submitted to this type of extraction, showed total and individual polyphenol values very similar to the control, with the exception of the higher content of caffeic acid, a phenolic acid well known for its many beneficial properties. In both cultivars, the supercritical extracts also showed a greater antioxidant activity than the control when measured by the ABTS^●+^ assay and a higher content of bergamotene and linalool, which is the main component of the essential oil of basil and is known for its bioactive properties, such as anti-inflammatory, antimicrobial, and antitumor activities. In conclusion, the basil leaf extracts obtained by supercritical CO_2_ and 10% of ethanol were rich in bioactive compounds and could be used in the food industry to prepare active packaging, in the cosmetic industry, or as pharmaceutical agents after appropriate validation.

## Figures and Tables

**Table 1 foods-11-03212-t001:** Yield (g/100 g DM) of extracts obtained from basil by different methods.

Code	Extraction Methods	Time (h)		Yield (g/100 g DM)
**C**	**Ethanol**	6		7.98 ± 0.25 ^a^
	**Supercritical CO_2_** (30 Mpa, 50 °C, 10 mL/min)		Ethanol (% *v*/*v*)	
**SC1**		5	0	0.58 ± 0.07 ^g^
**SC2**		5	5	5.10 ± 0.09 ^c^
**SC3**		5	10	6.54 ± 0.33 ^b^
**SC4**		2	0	<0.10
**SC5**		2	5	4.56 ± 0.20 ^c,d^
**SC6**		2	10	6.26 ± 0.33 ^b^
**SC7**		1	0	<0.10
**SC8**		1	5	3.91 ± 0.10 ^d,e^
**SC9**		1	10	5.32 ± 0.23 ^c^
	**Liquid CO_2_** (10 Mpa, 20 °C, 10 mL/min)		Ethanol (% *v*/*v*)	
**L1**		5	0	2.02 ± 0.11 ^f^
**L2**		5	5	4.34 ± 0.19 ^d^
**L3**		5	10	4.84 ± 0.06 ^c,d^
**L4**		2	0	<0.10
**L5**		2	5	3.44 ± 0.23 ^e^
**L6**		2	10	4.48 ± 0.11 ^d^
**L7**		1	0	<0.10
**L8**		1	5	2.39 ± 0.13 ^f^
**L9**		1	10	4.14 ± 0.07 ^d,e^

^a–g^: different letters in the same column indicate statistically significant differences (*p* ≤ 0.05).

**Table 2 foods-11-03212-t002:** Total phenol content (TPC), phenolic acids, and antioxidant activity determined by DPPH, ABTS, and FRAP assays in “Italiano Classico” and “Genovese” basil extracts obtained by different methods (C: ethanol; SC6: supercritical CO_2_ + 10% ethanol).

	“Italiano Classico” Basil		“Genovese” Basil	
	Extraction Methods		Extraction Methods	
	C	SC6	C	SC6
**TPC** (mg GAE/g extract)	91.66 ± 2.42 ^b^	72.15 ± 2.08 ^d^	98.99 ± 2.03 ^a^	79.62 ± 2.08 ^c^
**Phenolic acids** (mg/g extract)				
Rosmarinic acid	8.08 ± 0.26 ^b^	5.79 ± 0.18 ^c^	8.76 ± 0.24 ^a^	7.48 ± 0.18 ^b^
Caffeic acid	0.97 ± 0.04 ^b^	1.69 ± 0.04 ^a^	1.24 ± 0.06 ^b^	1.92 ± 0.05 ^a^
Chicoric acid	0.75 ± 0.02 ^a^	0.49 ± 0.02 ^b^	0.79 ± 0.02 ^a^	0.54 ± 0.02 ^b^
Ferulic acid	0.17 ± 0.01 ^a,b^	0.13 ± 0.01 ^c^	0.19 ± 0.01 ^a^	0.15 ± 0.01 ^b,c^
**Σ Phenolic acids**	10.07 ± 0.31 ^b^	8.20 ± 0.10 ^c^	10.78 ± 0.11 ^a^	10.19 ± 0.11 ^b^
**Antioxidant activity**				
**DPPH** EC_50_ (mg/mL)	1.32 ± 0.12 ^b^	1.86 ± 0.17 ^d^	1.14 ± 0.08 ^a^	1.52 ± 0.12 ^c^
**ABTS** EC_50_ (mg/mL)	1.29 ± 0.09 ^d^	0.84 ± 0.15 ^b^	1.11 ± 0.05 ^c^	0.73 ± 0.13 ^a^
**FRAP** (mmolTE/g extract)	0.90 ± 0.22 ^a^	0.75 ± 0.17 ^b^	0.92 ± 0.25 ^a^	0.801 ± 0.28 ^b^

^a–d^: different letters in the same row indicate statistically significant differences (*p ≤* 0.05).

**Table 3 foods-11-03212-t003:** Volatile organic compounds (relative percentage) in “Italiano Classico” and “Genovese” basil extracts obtained by different methods (C: ethanol; SC6: supercritical CO_2_ + 10% ethanol).

	“Italiano Classico” Basil		“Genovese” Basil	
	Extraction Methods		Extraction Methods	
	C	SC6	C	SC6
**Compound**				
**Σ Monoterpenes**	39.81 ± 1.30 ^a^	38.28 ± 1.54 ^a^	29.40 ± 1.45 ^c^	33.27 ± 1.06 ^b^
Eucalyptol	2.28 ± 0.36 ^b^	1.64 ± 0.75 ^b^	2.60 ± 0.18 ^b^	5.12 ± 1.81 ^a^
Linalool	31.51 ± 2.27 ^b^	35.08 ± 2.08 ^a^	23.29 ± 3.07 ^d^	27.17 ± 0.68 ^c^
Bornyl acetate	6.02 ± 0.62 ^a^	1.56 ± 0.52 ^bc^	3.51 ± 0.31 ^b^	0.98 ± 0.03 ^c^
**Monoterpenes ketones**				
Camphor	1.23 ± 0.02 ^a^	0.90 ± 0.07 ^a^	0.22 ± 0.08 ^c^	0.47 ± 0.01 ^b^
**Σ Monoterpenes alcohols**	1.75 ± 0.03 ^c^	1.58 ± 0.04 ^c^	4.95 ± 0.30 ^a^	1.97 ± 0.10 ^b^
α-Terpineol	0.77 ± 0.04 ^b^	0.87 ± 0.06 ^b^	2.16 ± 0.29 ^a^	0.93 ± 0.09 ^b^
Borneol	0.98 ± 0.03 ^b^	0.71 ± 0.05 ^c^	2.79 ± 0.38 ^a^	1.04 ± 0.25 ^b^
**Σ Sesquiterpenes**	27.16 ± 3.35 ^b^	32.76 ± 5.23 ^a^	23.77 ± 2.96 ^c^	32.67 ± 5.02 ^a^
β -Elemene	2.91 ± 0.40 ^b^	4.06 ± 0.11 ^a^	3.12 ± 0.41 ^b^	1.45 ± 0.33 ^c^
α-Guaiene	2.61 ± 0.38 ^a^	1.15 ± 0.27 ^b^	1.75 ± 0.19 ^ab^	0.88 ± 0.04 ^c^
Trans-α-Bergamotene	6.38 ± 0.43 ^d^	11.08 ± 0.81 ^b^	8.68 ± 0.82 ^c^	14.40 ± 1.20 ^a^
α-Cadinene	2.38 ± 0.19 ^a^	1.46 ± 0.20 ^b^	2.41 ± 0.12 ^a^	1.67 ± 0.40 ^b^
α-Caryophyllene	2.69 ± 0.26 ^a^	1.73 ± 0.15 ^ab^	1.26 ± 0.09 ^b^	1.79 ± 0.48 ^a,b^
D-Germacrene	0.97 ± 0.09 ^b^	1.03 ± 0.23 ^ab^	1.52 ± 0.26 ^a^	0.38 ± 0.02 ^c^
β-Cubebene	5.21 ± 0.56 ^a^	4.13 ± 1.29 ^a^	1.24 ± 0.14 ^b^	2.30 ± 1.19 ^b^
α-Cubebene	nd	0.92 ± 0.06 ^a^	0.34 ± 0.04 ^b^	0.10 ± 0.01 ^c^
γ-Elemene	1.30 ± 0.29 ^b^	2.09 ± 0.30 ^a^	0.46 ± 0.03 ^c^	0.55 ± 0.09 ^b,c^
Valencene	1.23 ± 0.25 ^ab^	1.52 ± 0.24 ^a^	0.48 ± 0.05 ^c^	1.01 ± 0.17 ^b^
γ-Muurolene	0.35 ± 0.08 ^c^	2.02 ± 0.60 ^a^	0.27 ± 0.04 ^c^	2.14 ± 0.42 ^b^
δ-Cadinene	0.54 ± 0.06 ^bc^	1.57 ± 0.37 ^a^	0.79 ± 0.09 ^b^	nd
γ-Selinene	0.59 ± 0.07 ^b^	nd	1.45 ± 0.22 ^a^	nd
**Allylbenzenes**				
Eugenol	27.27 ± 0.50 ^b^	25.42 ± 3.71 ^c^	38.92 ± 3.01 ^a^	35.78 ± 2.22 ^a^
**Alkylbenzenes**				
Methyl-isoeugenol	1.88 ± 0.10 ^b^	1.02 ± 0.14 ^b^	2.56 ± 0.76 ^a^	1.72 ± 0.38 ^ab^
**Σ Others**	0.90 ± 0.03 ^b^	0.04 ± 0.01 ^b^	0.18 ± 0.01 ^b^	0.12 ± 0.01 ^b^

^a–d^: different letters in the same row indicate statistically significant differences (*p ≤* 0.05). nd, not detected.

## Data Availability

Data is contained within the article.

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
