# Peer review of "Basil (Ocimum basilicum L.) Leaves as a Source of Bioactive Compounds"

_foods, 2022, doi:10.3390/foods11203212_

Round 1

Author Response

Thank you for the thoughtful comments and constructive suggestions, which helped to improve the quality of this manuscript. Our responses follow in red in the attached file.

Reviewer 2 Report

The article is well organized, written in good English style.

The methodology and the results are clearly presented, easy to read and well documented; the authors compared their results with other data previously published. The number of references is high enough to sustain the comparison of the results.

The only suggestion I have for the authors is to emphasize their original contribution (E.g. the use ethanol as a cosolvent).

Author Response

We would like to thank the reviewer for the careful and thorough reading of this manuscript.

We revised the point.  We emphasized the original contribution, in particular the use ethanol as a cosolvent.

Reviewer 3 Report

Section 2.4, needs proper reference of the method used.

Table 2, total phenolic acids should be added by summation of phenolic acids quantified and compare with TPC in discussion.

Author Response

Thank you for the careful and thorough reading of this manuscript.

We revised the paper according your suggestions.

In Section 2.4, we added Romano et al.

In Table 2 we added the sum of individual polyphenols and commented the that it showed the same trend of TPC. 

Reviewer 4 Report

The present manuscript deals with the study of the extraction of bioactive compounds from basil using supercritical CO2 and liquid CO2. The authors characterized the extraction of two basil cultivars by means of extraction yield, antioxidant activity, phenolic acid content and volatile organic compounds. The authors compare the results obtained with those provided by ethanol, used as a control.  

 The experiments carried out, are suitable for the aims of the manuscript and provide a comprehensive study of the extraction using liquid and supercritical CO2.

 Regarding the novelty of the manuscript, while there are some publications on this topic, this manuscript provides new details on the use of both supercritical CO2 and liquid CO2for basil extraction.

 In my opinion, the findings are interesting for a broader community and deserve to be published.

Despite its potential, the paper comes with a few issues which are addressed below:

·         Define abbreviations at first mention. Line 195 ABTS

·         In the tables 2 and 3, should have a short explanatory caption pointing out the meaning of C.

·         When using more than two correlative cites in text, put them like [26 – 29] instead of [26,27,28,29]. This can be found in line 55 and 106.

·         Revise the text, there are a few words misspelled:

o    Line 170 degaser should be degasser.

o    Line 171 d.i. should be i.d.

·         It is not wrong, but I would advise the authors to use the acronym DPO instead of DOP for protected designation of origin in Line 5.

 Best regards

Author Response

Thank you for the careful and thorough reading of this manuscript.

We revised the paper according your suggestions.

We defined abbreviations at first mention. Line 195 ABTS

In the tables 2 and 3 we added (C: ethanol; SC6: supercritical CO2 +10% ethanol) in the caption.

We revised correlative cites throughout the text.

We revise the misspelled words.

We revised the acronym PDO